# Repeatability, reproducibility, agreement, and safety of Tono-Pen tip cover for intraocular measurement using latex and polyethylene wrap

Pukkapol Suvannachart[1], Somkiat Asawaphureekorn[2], Sunee Chansangpetch[1], Abhibol Inobhas[1], Krit Pongpirul[3]*

1 Department of Ophthalmology, Glaucoma Research Unit, Faculty of Medicine, Chulalongkorn University, Bangkok, Thailand, 2 Department of Ophthalmology, Glaucoma Unit, Faculty of Medicine, Khon Kaen University, Khon Kaen, Thailand, 3 Department of Preventive and Social Medicine, Faculty of Medicine, Chulalongkorn University, Bangkok, Thailand

* doctorkrit@gmail.com

## Abstract

### Purpose

To evaluate repeatability, reproducibility, and agreement of intraocular pressure measurement with Tono-Pen using Ocufilm and polyethylene wrap tip cover in human eyes.

### Methods

This is a cross-sectional, experimental study. A gas-sterilized, polyethylene wrap was used as an alternative for Tono-Pen tip cover. For the right eye, 4 measurements using polyethylene wrap tip cover were done by two examiners (A and B) in random order to assess intra-observer repeatability and inter-observer reproducibility. For the left eye, 4 measurements were done by examiner A using both polyethylene wrap tip cover and Ocufilm in random order to assess intra-observer repeatability and agreement. Bland-Altman plot and intra-class correlation coefficient (ICC) were used in all analyses. Cost minimization analysis was evaluated.

### Results

For examiner A, the repeatability of polyethylene wrap tip cover was -0.34, 95% limits of agreement (LOA) were -3.04 to 2.36, and ICC was 0.93 in the right eyes. As for the left eyes, the repeatability of polyethylene wrap tip cover was -0.33, 95% LOA were -3.01 to 2.36, and ICC was 0.93. For examiner B, the repeatability of polyethylene wrap tip cover was -0.02, 95% LOA were -2.88 to 2.83, and ICC was 0.92. The inter-observer reproducibility of polyethylene wrap tip cover was 0.36, 95% LOA were -3.34 to 4.07, and ICC was 0.90. The repeatability of Ocufilm was -0.42, 95% LOA were -2.75 to 1.91, and ICC was 0.95. The agreement of polyethylene wrap tip cover and Ocufilm was -0.71, 95% LOA were -5.18 to 3.76, and ICC was 0.83. There were no allergic reactions or serious complications. From the

**Data Availability Statement:** All relevant data are within the manuscript and its Supporting Information files.

**Funding:** Funder Name: The 90th Anniversary of Chulalongkorn University (Rachadapisek Sompote Fund), and the Scholarship from the Graduate School, Chulalongkorn University to commemorate the 72nd anniversary of his Majesty King Bhumibala Aduladeja. Grant Recipient: Dr. Pukkapol Suvannachart The funder had no role in study design, data collection and analysis, decision to publish or preparation of the manuscript.

**Competing interests:** The authors have declared that no competing interests exist.

cost minimization analysis, the local cost for polyethylene tip cover was approximately 8 times lower compared to Ocufilm.

## Conclusions

Tono-pen with Ocufilm and polyethylene wrap tip cover were used to measure the intraocular pressure. The polyethylene wrap tip cover demonstrated acceptable repeatability, reproducibility, and agreement with Ocufilm in normotensive eyes, and had a good safety profile.

## Introduction

One of the most important procedure in routine ocular examination is the measurement of the intraocular pressure (IOP). Indirect IOP measurement using different techniques, such as applanation and indentation, are generally used in clinics. Tono-Pen® (TP; Reichert, New York, USA) is a handheld, digital tonometer that involves both indentation and applanation mechanisms. It demonstrates comparable measurement of IOP compared to the Goldmann applanation tonometry (GAT) which is the gold standard method [1, 2]. It can measure the IOP in a small area on the cornea and can be used on patient in any position. This easy-to-use device can be used by novice medical personnel without compromising the result [3]. There is a good intra-session repeatability in both glaucoma and healthy patients [4]. The use of TP requires a tip cover to prevent damage to the transducer tip and cross contamination. Ocufilm® (OF, Reichert, New York, USA) is a commercially available disposable tip cover made from latex that can cause allergy. The prevalence of latex allergy in the general population worldwide is approximately 4.3% [5]. Patients with severe latex allergy were reported to develop conjunctival injection, eyelid erythema, and eyelid edema after IOP measurement with latex tip cover [6].

OF tip cover is sanitized but not sterilized. This prevents its use in post-operative eyes that need sterile instrument for IOP measurement. The cost of this single use tip cover may cause financial burden, especially in developing countries. In addition, sometimes there are shortages of OF tip cover. In order to overcome these barriers, a previous study demonstrated that a fingertip of the surgical glove could be used as a tip cover and showed satisfactory repeatability and agreement with OF [7]. However, the cost of latex surgical glove is significant and latex allergy is still possible.

Alternative material should be considered such as plastic wrap for packaging the food which has a smooth surface and barrier properties against moisture, gas, and organisms. This economical and readily available material can be attached to any surface without adhesive. The majority of food grade plastic wrap is made from either polyethylene or polyvinyl chloride. Both materials can withstand heat up to 120–130 degrees Celsius so they can be gas sterilized. Both types of the plastics are widely used in medical applications, such as catheters and synthetic materials. In ophthalmology, polyethylene has long been used in ocular surgery [8]. Its use can cause some postoperative reaction after being inserted into rabbit eyes [9]. Plastic wrap has been reported to be used as a barrier in GAT and contact A-scan ultrasonography [10–12]. For IOP measurement with TP, our previous eye model and study conducted in canine eyes showed good repeatability and agreement between the custom-made polyethylene wrap (PW) tip cover and OF without causing any ocular surface complications [13, 14].

The purpose of this study was to evaluate the repeatability, reproducibility, and agreement of IOP measurement with TP using OF and PW in human eyes. The safety and cost comparison between both tip covers were also evaluated.

## Materials and methods

This is a cross-sectional, experimental study. It was approved by the institutional review board of the Faculty of medicine, Chulalongkorn University, Bangkok, Thailand, and adhered to the tenets of the Declaration of Helsinki. This trial was registered in the Thai Clinical Trial Registry (TCTR) and its identification number is TCTR20190108001. Written informed consent was obtained from all participants.

### Participants

Ophthalmic patients and healthy volunteers, at least 18 years old, were invited to enroll in this study. All participants underwent a thorough ocular examination, including visual acuity testing, and slit lamp examination of the anterior and posterior segments. Those with a history of plastic or latex allergy, history of intraocular surgery rather than an uncomplicated small incision cataract surgery with phacoemulsification technique, or have corneal surface pathologies in either eye such as abrasion, infiltration, and scar, were excluded from the study. IOP measurement was done by two examiners; examiner A was an ophthalmologist and examiner B was a general practitioner.

### IOP measurement

Tono-Pen AVIA® was used in this study. Although regular calibration is not necessary, it was performed per standard protocol recommended by the manufacturer once at the beginning of the day. A drop of 0.5% tetracaine was instilled to both eyes to achieve adequate anaesthesia. Ten gentle applanations at the central cornea were performed for each measurement to obtain an average IOP. Only IOP reading with a statistical confidence indicator of 95 was considered reliable and was used in the analysis.

### PW tip cover

PW used in this study (Cleanwrap®, Seoul, Korea) which was identical to that used in our previous studies [13, 14]. The thickness of the film was 10 micrometres. It was sent for cytotoxic testing at the National Metal and Materials Technology Center (MTEC) using 3-(4,5-dimethylthiazol-2-yl)-2,5-diphenyltetrazolium bromide (MTT) assay which tested the viability of mouse fibroblast after 24-hour exposure to the material. The test demonstrated no cytotoxic potential with 100% viability of the cells (report no. MTEC1200/62).

PW tip cover was prepared by cutting the commercial PW to a size of 5 by 5 centimetres. A cover holder was created by cutting the paper ring from the OF tip cover package in half widthwise (Fig 1A). Both materials were put in a sterilization pouch. This package was sent for ethylene oxide sterilization (Fig 1B). When in use, the sterilized pouch was peeled off and the PW was carefully lifted out of the pouch by touching only its edge without touching the central part of the PW. Then, the PW was placed over the TP tip and the cover holder was put on top of the PW and advanced to the neck of the TP head until the PW was secured in place and attached smoothly over the TP tip (Fig 2B).

### Measurement protocol

The overall study flow diagram is shown in Fig 3. The right eye of each participant was used to assess the intra-observer repeatability and inter-observer reproducibility of PW. Four IOP measurements, performed twice by examiners A and B, were done on the right eye.

The left eye of each participant was used to study the agreement between OF and PW tip covers and intra-observer repeatability of both types. Four IOP measurements, twice for PW

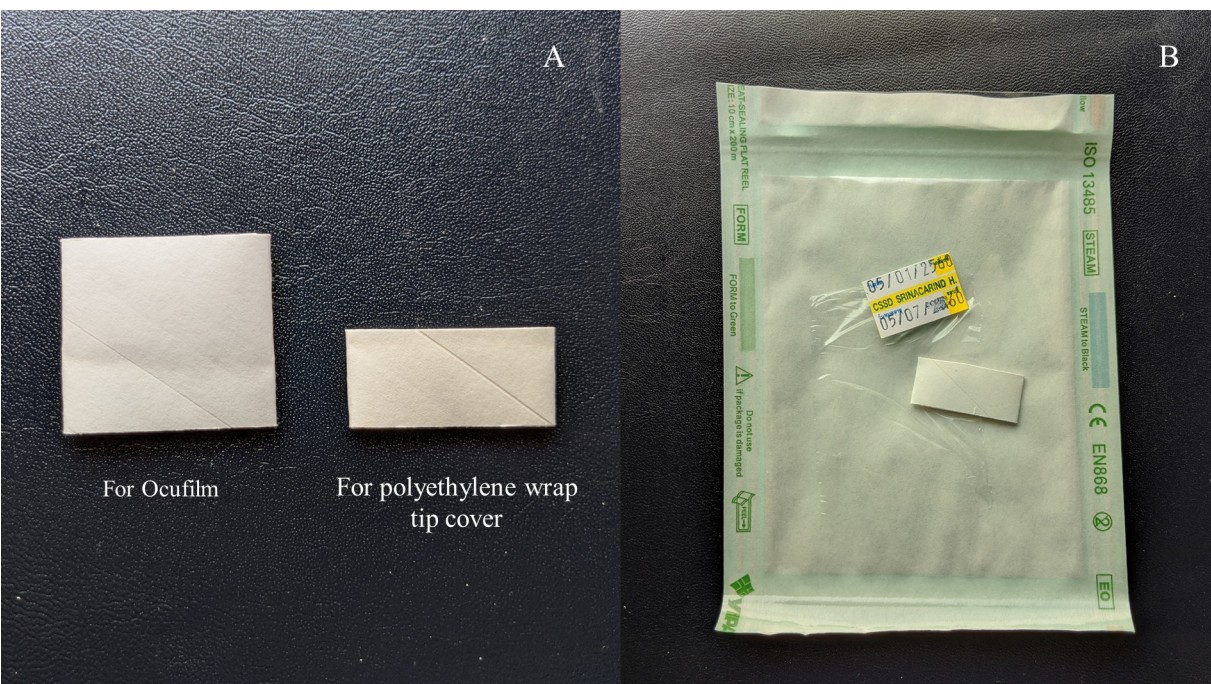

**Fig 1. Polyethylene wrap tip cover preparation** (A) Cover holders, (B) Polyethylene wrap tip cover package.

and OF, were done by examiner A. To balance the effect of decreasing IOP after repeated measurements, the order of measurement by the tip covers and the examiners were done in random sequence.

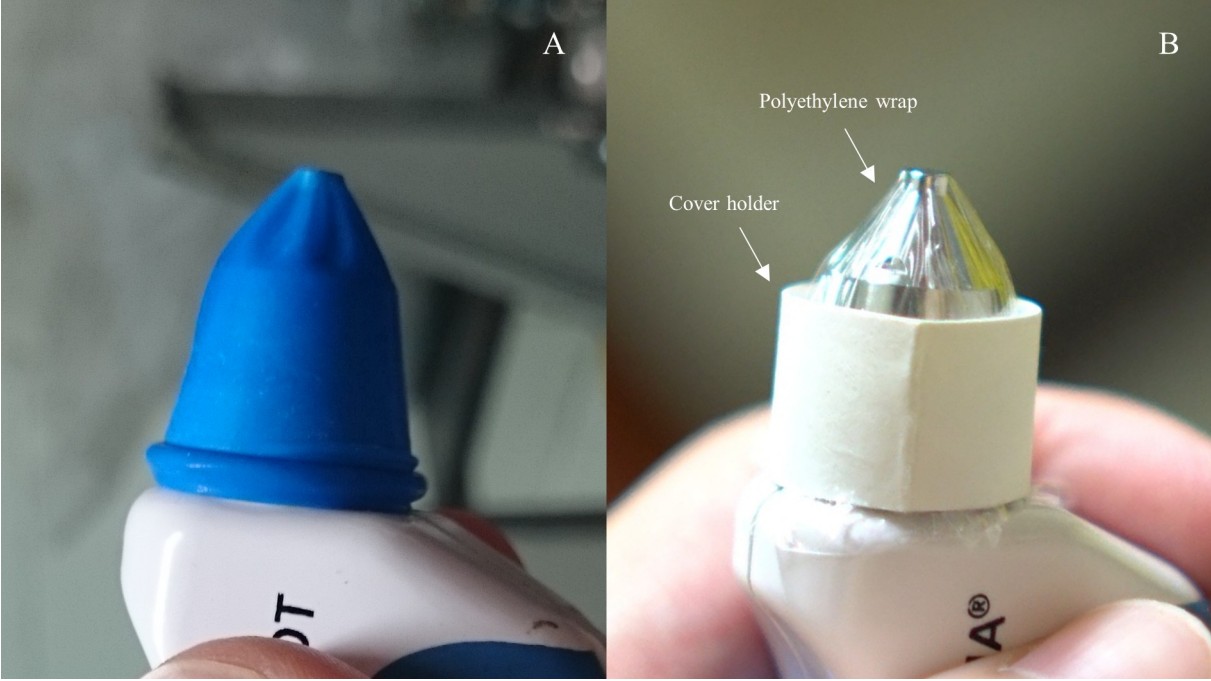

**Fig 2. Tono-Pen tip covers.** (A) Ocufilm, (B) Polyethylene wrap with the cover holder in place.

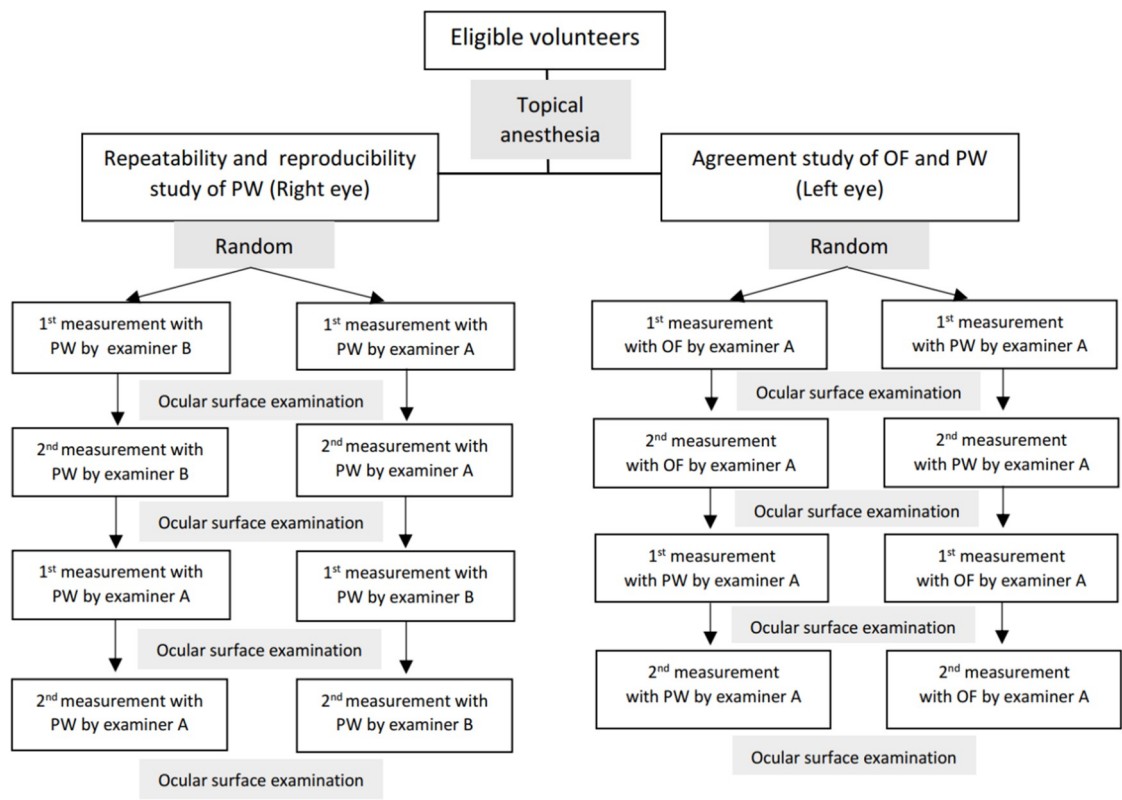

**Fig 3. The study flow diagram.**

For safety evaluation, ocular surface examination with fluorescein staining under cobalt blue light was performed after each measurement to detect any damages that might have occurred during the procedure. Any new pathologies compared to the baseline examination, including punctate epithelial erosion, corneal epithelial defect, conjunctival injection, conjunctival papillary reaction, and chemosis, were noted.

## Statistical analyses

All statistical analyses and graphs were generated using MedCalc for Windows, version 19.2 (MedCalc Software, Ostend, Belgium). The unit of analysis was the eye. Baseline characteristics and complications were reported using descriptive statistics as appropriate. For intra-observer repeatability of OF and PW, Bland-Altman plot (BA plot) was used to demonstrate mean differences (MD), limits of agreement (LOA) and their 95% confidence intervals (95% CI) [15]. For inter-observer reproducibility and agreement between OF and PW, BA plot with multiple measurements per participant was used [16]. Intraclass correlation coefficient (ICC) estimates and their 95% CI were calculated based on the following parameters: mean rating (k = 2), absolute agreement, two-way model, single measures and same raters for all participants. ICC estimates were interpreted using Koo and Li classification [17]. Cost minimization analysis was performed by comparing the cost of materials and production of each tip cover.

## Results

A total of 128 participants (256 eyes) were recruited into the study. Majority of the participants were female (78.1%). The mean age ± standard deviation was 46.0 ± 16.6 years (range 18–83). Nine of the right eyes and 4 of the left eyes had pseudophakia.

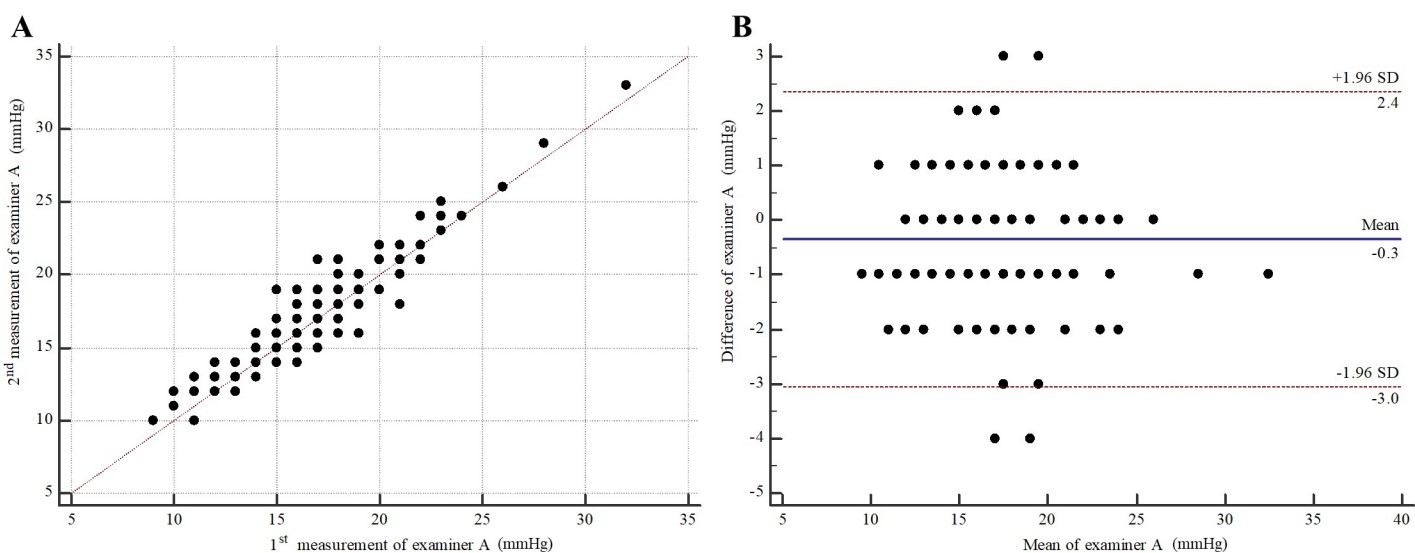

**Fig 4. Intra-observer repeatability of polyethylene wrap tip cover by examiner A (right eyes).** (A) Scatter plot. (B) Bland-Altman plot.

For intra-observer repeatability of PW by examiner A (right eyes), the MD (95% CI) was -0.34 (-0.58 to -0.10). The 95% LOA (95% CI) were -3.04 (-3.46 to -2.63) to 2.36 (1.94 to 2.77) (Fig 4). The ICC (95% CI) was 0.93 (0.90 to 0.95). For intra-observer repeatability of PW by examiner B (right eyes), the MD (95% CI) was -0.02 (-0.28 to 0.23). The 95% LOA (95% CI) were -2.88 (-3.31 to -2.44) to 2.83 (2.39 to 3.26) (Fig 5). ICC (95% CI) was 0.92 (0.89 to 0.94). For inter-observer reproducibility, the MD (95% CI) between examiner A and examiner B was 0.36 (0.09 to 0.64). The 95% LOA (95% CI) were -3.34 (-3.84 to -2.93) to 4.07 (3.66 to 4.56) (Fig 6). The ICC (95% CI) was 0.90 (0.85 to 0.93).

For intra-observer repeatability of OF by examiner A (left eyes), the MD (95% CI) was -0.42 (-0.63 to -0.21). The 95% LOA (95% CI) were -2.75 (-3.11 to -2.39) to 1.91 (1.55 to 2.26) (Fig 7). ICC (95% CI) was 0.95 (0.92 to 0.97). For intra-observer repeatability of PW by examiner A

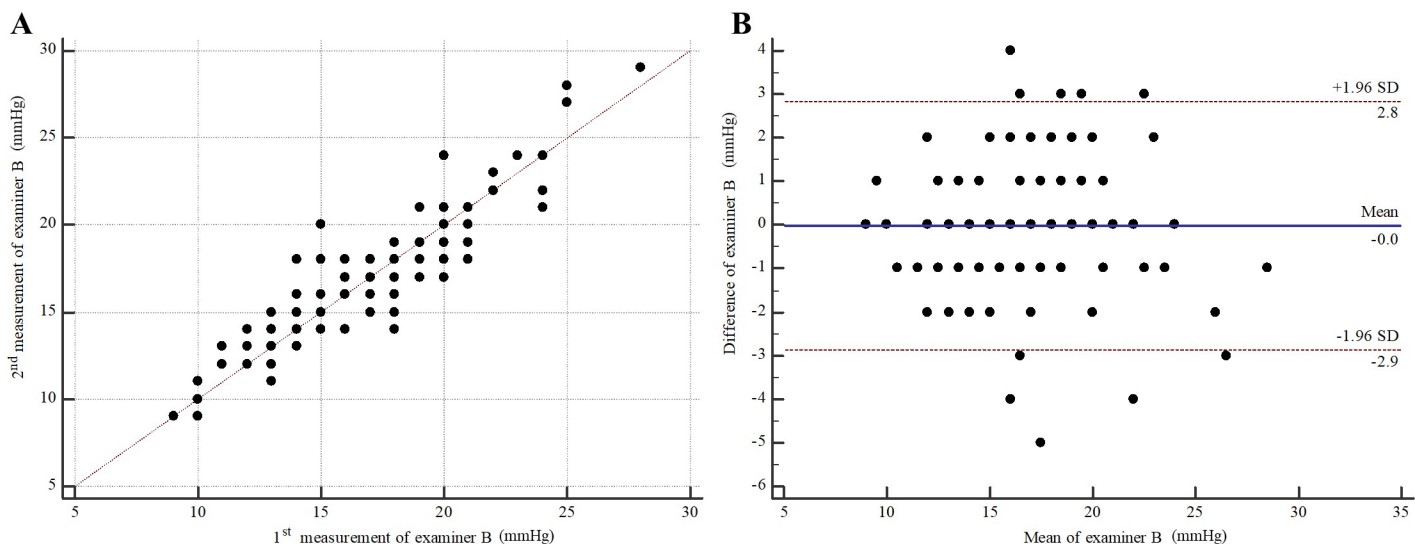

**Fig 5. Intra-observer repeatability of polyethylene wrap tip cover by examiner B (right eyes).** (A) Scatter plot. (B) Bland-Altman plot.

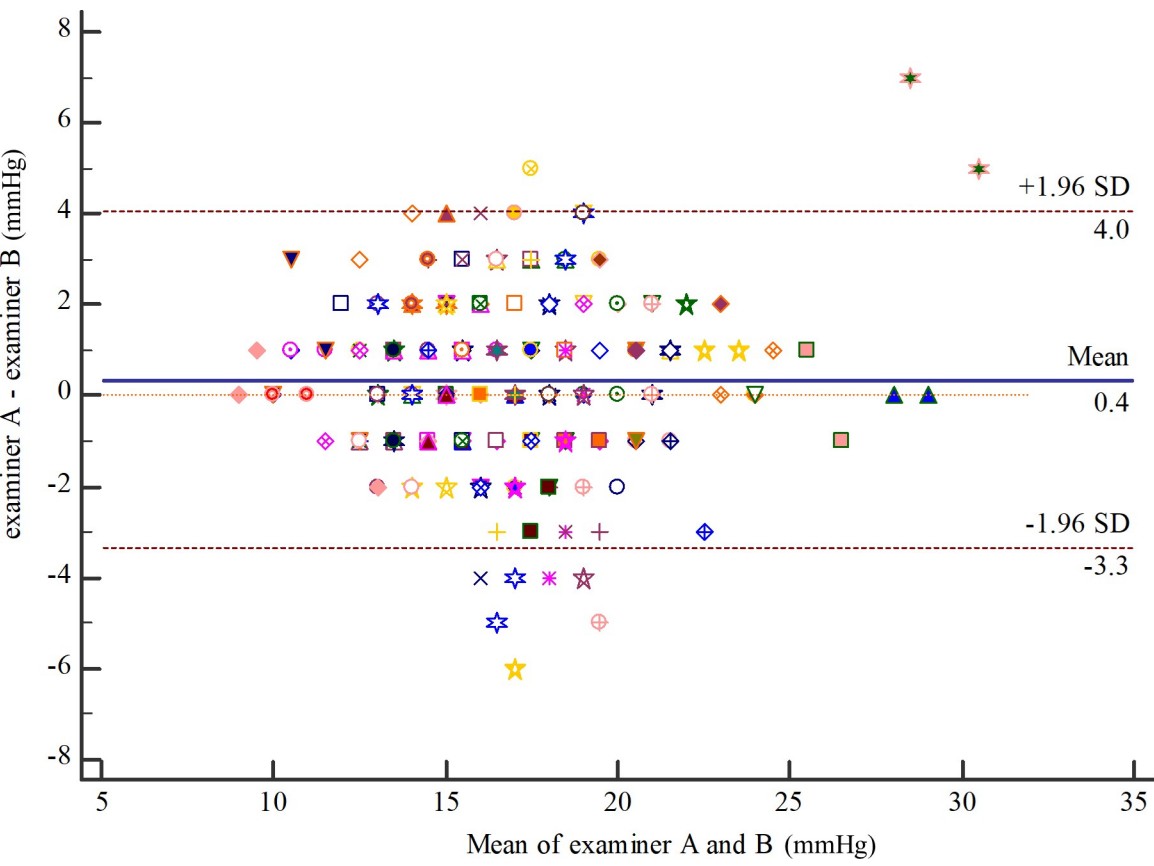

**Fig 6. Inter-observer reproducibility of polyethylene wrap tip cover between examiners A and B (right eyes).**

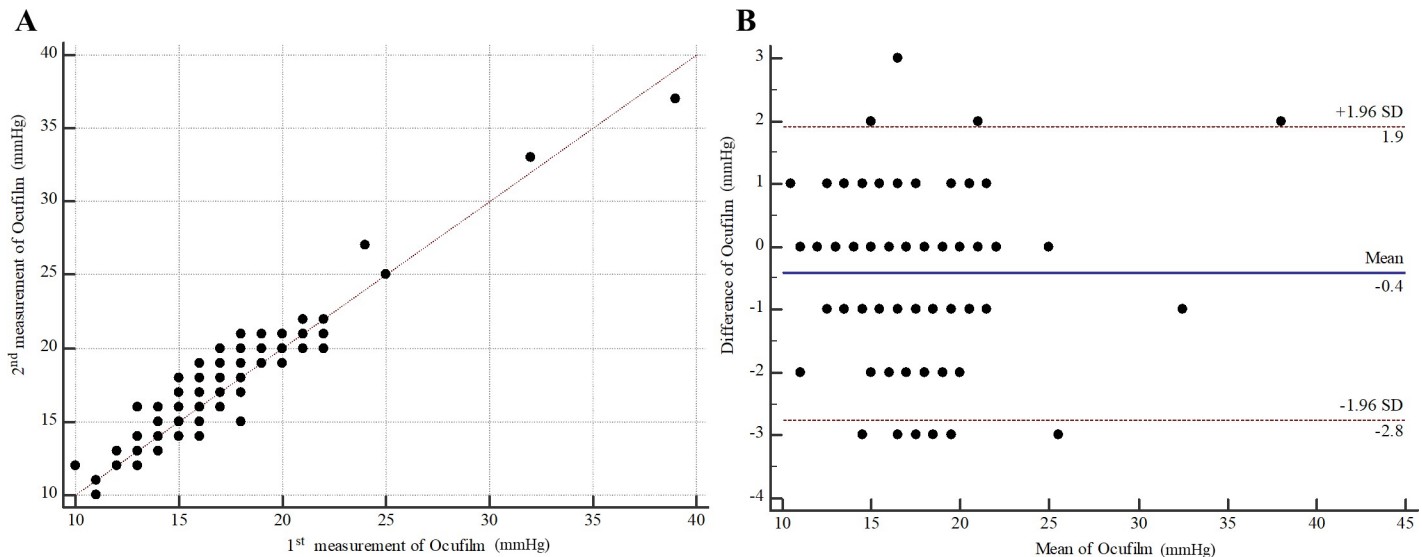

**Fig 7. Intra-observer repeatability of Ocufilm by examiner A (left eyes).** (A) Scatter plot. (B) Bland-Altman plot.

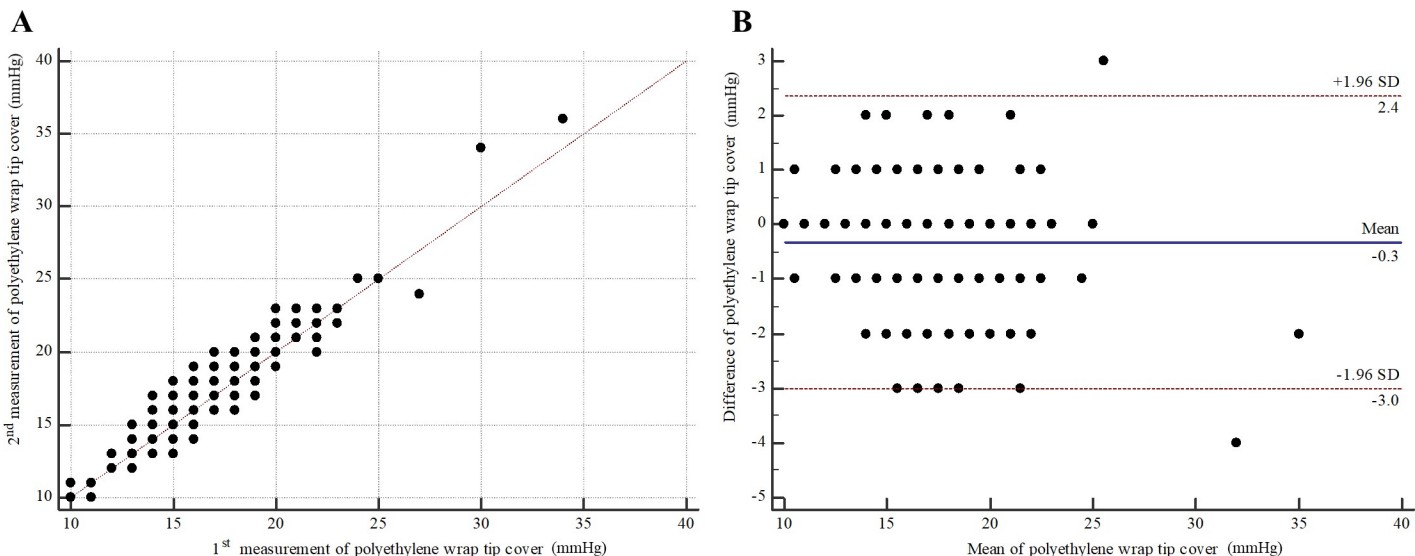

**Fig 8. Intra-observer repeatability of polyethylene wrap tip cover by examiner A (left eyes).** (A) Scatter plot. (B) Bland-Altman plot.

(left eyes), the MD (95% CI) was -0.33 (-0.57 to -0.09). The 95% LOA (95% CI) were -3.01 (-3.42 to -2.60) to 2.36 (1.95 to 2.77) (Fig 8). ICC (95% CI) was 0.93 (0.90 to 0.95). For agreement between OF and PW by examiner A (left eyes), the MD (95% CI) was -0.71 (-1.07, -0.35). The 95% LOA (95% CI) were -5.18 (-5.83 to -4.63) to 3.76 (3.21 to 4.40) (Fig 9). ICC (95% CI) was 0.83 (0.75 to 0.89). All analyses are summarized in Table 1.

All of the eyes had no serious complications post-measurement by both OF and PW. The only complication found in the study was punctate epithelial erosion (PEE). For PW, 7 (5.5%) of the right eyes and 1 (0.8%) of the left eye had PEE after all measurements were done. For OF, 3 (2.3%) of the left eyes were found to have PEE. None of the participants reported any post-measurement complications within the first 24 hours. There were no vision-threatening complications such as corneal epithelial defect and keratitis, or any allergic reactions.

Table 2 shows the detail of the cost to produce PW. From the cost minimization analysis, the average cost of OF was around 0.8 USD. The average cost of one PW was 0.1 USD. The cost difference between OF and PW tip cover was 0.7 USD. All costs were calculated from local purchase with local currency (Thai Baht). The exchange rate at the time of this study was around 32 Baht per 1 USD.

## Discussion

In the present study, we have demonstrated that PW could be used as an alternative to Ocufilm for TP tip cover. Both PW and OF had comparable intra-observer repeatability and inter-observer reproducibility with good agreement. According to published literature, test-retest variability of OF was 0.1 mmHg, 95% LOA were -3.3 to 3.5 mmHg, and the ICC range was 0.82 to 0.85 [2, 18, 19]. In our study, the intra-observer repeatability of OF was -0.42 mmHg but the LOA were narrower (-2.75 to 1.91 mmHg) with excellent ICC (0.95) (Table 1). The intra-observer repeatability of PW by both examiner A (right eye = -0.34 mmHg, left eye = -0.33 mmHg) and examiner B (right eye = -0.02 mmHg) were acceptable. The LOA of examiner A (right eye = -3.04 to 2.36 mmHg, left eye = -3.01 to 2.36 mmHg) and examiner B (right eye = -2.88 to 2.83 mmHg) were acceptable. Both examiners produced very similar LOA which indicated that the IOP measurement with PW was independent of the examiner's

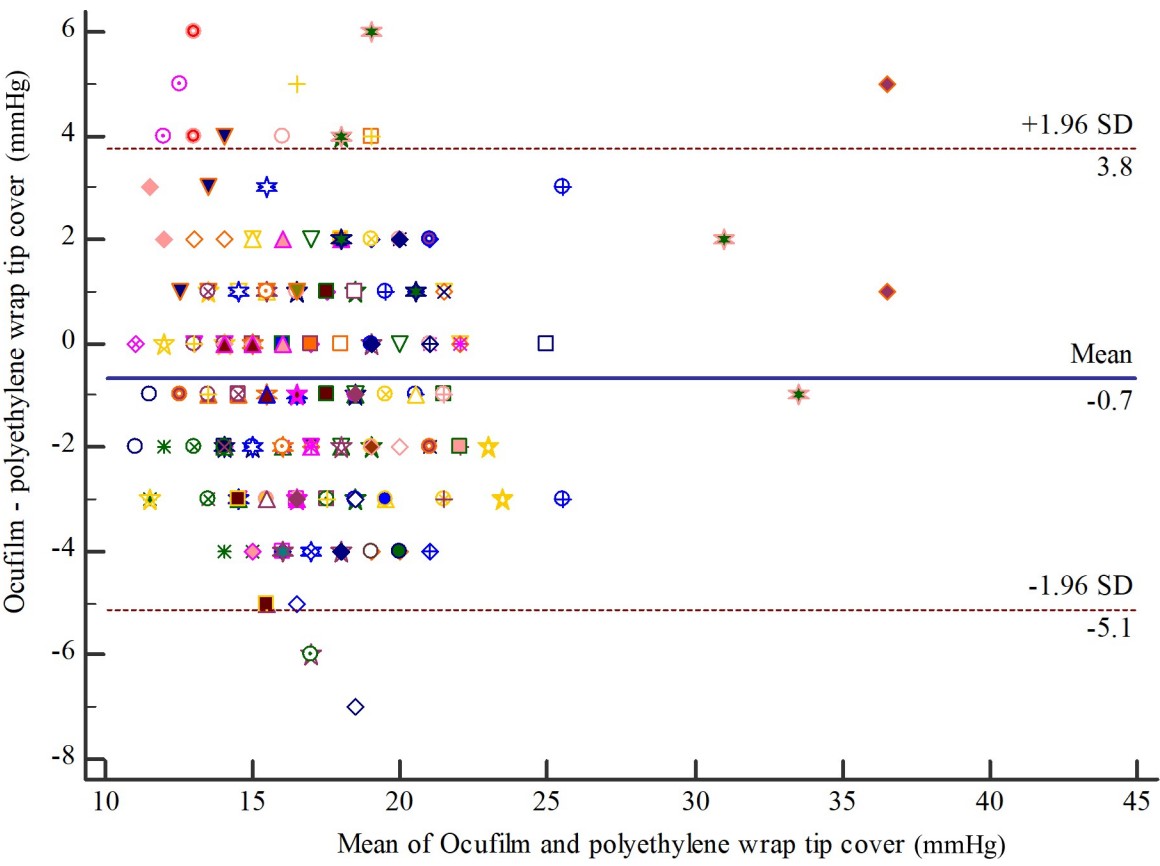

**Fig 9. Bland-Altman plot of agreement between Ocufilm and polyethylene wrap tip cover by examiner A (left eyes).**

experience. All ICC were classified as good to excellent. These results were similar to our previous studies that used PW in an eye model (intra-observer repeatability = -0.25 mmHg, 95% LOA = -4.55 to 4.05 mmHg) [13] and in canine eyes (intra-observer repeatability = 0.27 mmHg, 95% LOA = -2.74 to 3.27 mmHg) [14].

The inter-observer reproducibility of PW (0.36 mmHg) and LOA (-3.34 to 4.07 mmHg) were acceptable (Table 1) and comparable to our previous study done in canine eyes (inter-observer reproducibility = -0.39 mmHg, 95% LOA = -4.79 to 4.01 mmHg) [14].

PW and OF are different in many aspects, including materials and thickness. The thickness of PW is 10 microns. OF is grossly thicker, but its actual thickness is not available from the manufacturer. However, our study found an acceptable agreement between both tip covers, although PW produced slightly higher readings. In addition, the ICC showed moderate to good agreement. Similar results were found in our previous eye model (MD = 0.29, 95% LOA = -5.68 to 6.26) [13] and study conducted in canine eyes (MD = 0.13, 95% LOA = -3.92 to 4.18) [14]. Thus, PW and OF could be used as TP tip cover interchangeably.

It is interesting that TP itself has a relatively wide range of 95% LOA as previously mentioned, and the range is even larger when it is compared to other tonometers. Compared to GAT, the current gold standard, TP demonstrated varying results in different populations, with MD varying from -0.27 to 1.63 mmHg and the range of 95% LOA were 4.88 to 16.34 mmHg in normal individuals [1, 18, 20–23]. For glaucoma patients, especially those who have uncontrolled and elevated IOP, MD could be as high as 6.8 (range of 95% LOA = 25.9 mmHg) [24]. A systematic review denoted that only 48% of the results from TP were within 2 mmHg

**Table 1. Summary of Bland-Altman analyses and intraclass correlation coefficients.**

| Analysis | Mean difference (95% CI) mmHg | 95% limits of agreement (mmHg) | | ICC |
|---|---|---|---|---|
| | | Lower (95% CI) | Upper (95% CI) | (95% CI) |
| Right eyes | | | | |
| Intra-observer repeatability (A) | -0.34 (-0.58, -0.10) | -3.04 (-3.46, -2.63) | 2.36 (1.94, 2.77) | 0.93 (0.90, 0.95) |
| Intra-observer repeatability (B) | -0.02 (-0.28 to 0.23) | -2.88 (-3.31, -2.44) | 2.83 (2.39, 3.26) | 0.92 (0.89, 0.94) |
| Inter-observer reproducibility | 0.36 (0.09, 0.64) | -3.34 (-3.84, -2.93) | 4.07 (3.66, 4.56) | 0.90 (0.85, 0.93) |
| Left eyes | | | | |
| Intra-observer repeatability of OF (A) | -0.42 (-0.63, -0.21) | -2.75 (-3.11, -2.39) | 1.91 (1.55, 2.26) | 0.95 (0.92, 0.97) |
| Intra-observer repeatability of PW (A) | -0.33 (-0.57, -0.09) | -3.01 (-3.42, -2.60) | 2.36 (1.95, 2.77) | 0.93 (0.90, 0.95) |
| Agreement between OF and PW (A) | -0.71 (-1.07, -0.35) | -5.18 (-5.83, -4.63) | 3.76 (3.21, 4.40) | 0.83 (0.75, 0.89) |

Abbreviations: CI = confidence interval; ICC = intraclass correlation coefficient; OF = Ocufilm tip cover; PW = polyethylene wrap tip cover; A = examiner A; B = examiner B.

from GAT value [25]. This high variation could partly be explained by the dynamic change of IOP under several factors such as time of measurement, repeated measurements, area of cornea contact, patient's stress and unintentional Valsalva maneuver, and the examiner's experience. Another reason is that TP measures IOP instantaneously and has a very short contact time resulting in a greater variation of IOP especially in patients with a wider ocular pulse pressure [26].

In terms of safety, in this study, there were no sight-threatening complications and allergic reaction for both OF and PW. There were only a few participants from PW and OF groups that developed PEE post-measurements. This might be explained by the mechanical damage from multiple measurements. Furthermore, the contact area of TP was small (2.36 mm$^2$) compared to GAT (7.35 mm$^2$), and the contact time was very short. None of the participants with PEE required revisiting or further treatment. Thus, PW was safe for IOP measurement.

PW has many advantages. The material is generally available. PW tip cover can be easily prepared in any hospital with a significant cost reduction compared to the commercial product. The cost minimization analysis found that the cost of PW was approximately 8-times lower than the cost for OF. Another advantage of using PW over OF is that it can be gas sterilized. Consequently, PW can be used in post-operative patients where sterility is of concern. The sterilization process with ethylene oxide is commonly used and has high efficiency in eradicating microorganisms without deleterious effects on the plastic material [27]. In addition, compared to OF, PW can be used safely in patients with latex allergy. This method may also be suitable for a situation of high demand of use due to a concern of cross contamination like the recent outbreak of coronavirus disease 2019 (COVID-19).

**Table 2. Production cost of polyethylene wrap tip cover.**

| Items | Cost (Baht per unit) |
|---|---|
| Polyethylene wrap cost | 0.05 |
| Cover holder cost* | - |
| Packaging and gas sterilization cost | 2 |
| Labour cost | 1 |
| Total production cost | 3.05 |

*Derived from Ocufilm package.

There were some limitations in this study. Firstly, most of the participants had IOP within normal range. TP has lower accuracy when the patients have an extreme IOP [28]. This should be considered prior to using PW in clinical practice. In addition, our previous study found greater MD and wider LOA in the eye model with higher IOP range compared to the lower ones [13]. Future studies are required to validate the agreement between PF and OF tip cover in eyes with higher IOP range. Secondly, the efficacy of TP has high variation, thus, it would be better to compare the performance of both tip covers against another reliable instrument with less variability such as GAT and manometer. Furthermore, the central corneal thickness (CCT) was not measured in our study because we made a comparison within the same eye of the individual. However, the difference in CCT might affect the accuracy of the measurement [29]. We also did not measure the PW thickness after gas sterilization. The thickness may have been altered and become uneven after sterilization which could affect the IOP measurement.

## Conclusions

PW tip cover was safe and demonstrated acceptable intra-observer repeatability, inter-observer reproducibility, with good agreement compared to OF tip cover for IOP measurement with Tono-Pen in normotensive eyes. PW could be used as an alternative tip cover for Tono-Pen.

## Supporting information

**S1 File. Certification letter for English editing.**
(PDF)

**S1 Data.**
(XLSX)

## Author Contributions

**Conceptualization:** Pukkapol Suvannachart, Somkiat Asawaphureekorn.

**Data curation:** Pukkapol Suvannachart, Sunee Chansangpetch, Abhibol Inobhas.

**Formal analysis:** Pukkapol Suvannachart, Somkiat Asawaphureekorn.

**Funding acquisition:** Pukkapol Suvannachart, Sunee Chansangpetch, Krit Pongpirul.

**Investigation:** Pukkapol Suvannachart, Sunee Chansangpetch, Abhibol Inobhas.

**Methodology:** Pukkapol Suvannachart, Somkiat Asawaphureekorn, Sunee Chansangpetch, Krit Pongpirul.

**Project administration:** Pukkapol Suvannachart, Sunee Chansangpetch, Krit Pongpirul.

**Resources:** Pukkapol Suvannachart, Sunee Chansangpetch, Abhibol Inobhas.

**Supervision:** Somkiat Asawaphureekorn, Sunee Chansangpetch, Krit Pongpirul.

**Validation:** Somkiat Asawaphureekorn, Sunee Chansangpetch, Krit Pongpirul.

**Visualization:** Pukkapol Suvannachart.

**Writing – original draft:** Pukkapol Suvannachart, Abhibol Inobhas.

**Writing – review & editing:** Somkiat Asawaphureekorn, Sunee Chansangpetch, Krit Pongpirul.

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
