## [Decision Letter · Decision Letter 0]

10 Jul 2020

PONE-D-20-15592

Repeatability, reproducibility, agreement, and safety of Tono-Pen tip cover for intraocular measurement using latex and polyethylene wrap.

PLOS ONE

Dear Dr. Pongpirul,

Thank you for submitting your manuscript to PLOS ONE. After careful consideration, we feel that it has merit but does not fully meet PLOS ONE’s publication criteria as it currently stands. Therefore, we invite you to submit a revised version of the manuscript that addresses the points raised during the review process.

Both reviewers have raised a number of constructive criticisms that need to be carefully addressed in revising the manuscript. 

We look forward to receiving your revised manuscript.

Kind regards,

Sanjoy Bhattacharya

Academic Editor

PLOS ONE

Journal Requirements:

Reviewers' comments:

Reviewer's Responses to Questions

**Comments to the Author**

1. Is the manuscript technically sound, and do the data support the conclusions?

Reviewer #1: Partly

Reviewer #2: Yes

2. Has the statistical analysis been performed appropriately and rigorously? 

Reviewer #1: Yes

Reviewer #2: Yes

3. Have the authors made all data underlying the findings in their manuscript fully available?

Reviewer #1: Yes

Reviewer #2: Yes

4. Is the manuscript presented in an intelligible fashion and written in standard English?

Reviewer #1: No

Reviewer #2: Yes

5. Review Comments to the Author

Reviewer #1: GENERAL:

This manuscript describes a study in which the repeatability, reproducibility, and agreement of intraocular pressure (IOP) measurement with the Tono-Pen AVIA, a portable McKay-Marg tonometer, using Ocufilm and polyethylene wrap tip covers in human eyes was evaluated.

A gas-sterilized, polyethylene wrap was used as an alternative to the Tono-Pen tip cover as supplied by the manufacturer.

To assess intra-observer repeatability and inter-observer reproducibility, in the right eyes of human subjects, 4 IOP measurements using a polyethylene wrap tip cover were performed by two examiners, A and B, in a randomized manner.

In order to assess intra-observer repeatability and agreement, in the left eyes of human subjects, 4 IOP measurements were performed by two examiners, A and B, using both a polyethylene wrap tip cover and the Ocufilm tip cover in a randomized manner.

Bland-Altman plot and intra-class correlation coefficients (ICC) were used in all analyses. Cost minimization analysis was also performed.

For examiner A, the repeatability of polyethylene wrap tip cover = -0.34, 95% limits of agreement (LOA) = -3.04 to 2.36, and ICC = 0.93 in the right eyes, and -0.33, 95% LOA = -3.01 to 2.36, and ICC = 0.93 in the left eyes.

For examiner B, the repeatability of polyethylene wrap tip cover = - 0.02, 95% LOA = -2.88 to 2.83, and ICC = 0.92. The inter-observer reproducibility of polyethylene wrap tip cover = 0.36, 95% LOA = -3.34 to 4.07, and ICC = 0.95. The repeatability of Ocufilm = - 0.42, 95% LOA =-2.75 to 1.91, and ICC = 0.95.

The authors conclude that for intraocular pressure measurement with the Tono-Pen AVIA, the polyethylene wrap tip cover demonstrates acceptable repeatability, reproducibility, and agreement with the Ocufilm tip cover, and exhibits a good safety profile, at about 1/8 × the cost.

SPECIFIC COMMENTS:

Study Design seems reasonably sound. However a major weakness was, as the authors acknowledge, only ocular normotensive eyes were studied. The essential conclusion of the study, that the Ocufilm versus the polyethylene cover make no significant difference to IOP measurements, may be altered if ocular hypertensive eyes were to be studied. This needs to be addressed, as expounded below.

English is quite good but still needs to be redone by a native speaker or professional English writing service. There are too many grammatical errors.

How was the Tono-Pen AVIA calibrated? Please explain in detail.

In the IOP measurement section, it is stated that ten IOP measurements were taken, and then a mean value was calculated. Was this mean value considered to be a single IOP reading, such that, if 4 IOP values are stated as having been made, does this then imply 4 x 10 = 40 readings, i.e. 40 individual tonometer-corneal contacts?

The authors make mention of pseudofacility (i.e. reduction in IOP with repeated tonometric measurement, likely if a large number of individual readings were obtained using this type of device), but make no attempt to indicate by how much IOP may have diminished as a consequence of this. Can they give some indication of this?

Units (mmHg) should be included on axes of graphs.

Where is the Legend for Figure 6?

If I am reading it correctly, Figure 6 implies that there was a difference of as much as 4 mmHg between measurements made by Examiner A versus Examiner B. Isn’t this a rather high degree of variation? Or is this simply variation due to pulse amplitude?

Again, if I am reading it correctly, Figure 9 implies that there was a difference of as much as 6 mmHg between measurements made with the OF versus the PW tip covers. Isn’t this also a rather a high degree of variation? Or again, is this simply variation due to pulse amplitude?

As mentioned above, one weakness of the study, as the authors state, is that only ocular normotensive eyes were studied. The Tono-Pen tends to underestimate IOP in ocular hypertensive eyes. It very well might be the case that the different covers (Ocufilm versus polyethylene) could lead to a bigger difference in IOP measurements (depending on how high the true IOP actually is), perhaps too big to be clinically useful. The authors do state this but this needs to be made very clear. Is there any animal data, for example, in their dog study, that might show that the two different covers do not make a significant difference in the tonometric readings at high IOPs? Or any other published data. It is really necessary to show this, prior to using the tonometer with a polyethylene cover in a clinical setting. Perhaps a similar study could be performed in ocular hypertensive patients, after comparing Tono-Pen data with corresponding GAT data.

When the authors mention their measurements of IOP in dogs in their previous study, did they use the Tono-Pen AVIA? Or the Tono-Pen AVIA Vet (a similar but differently calibrated device, intended for use not with humans but with dogs and cats). Or a different version of the Tono-Pen?

Reviewer #2: Will there be any consideration to adjust stay design to mitigate the risk of Punctate Epithelial Erosions?

Line 4: "It demonstrated" changed to "It Demonstrates"

Line 6: Change to "Can be used on patient in any position"

Remove "sometimes happens" from the end of "In addition to shortage at supply"

6. PLOS authors have the option to publish the peer review history of their article (what does this mean?). If published, this will include your full peer review and any attached files.

Reviewer #1: No

Reviewer #2: **Yes: **Mike Zein, MD

---

## [Author Response · Author response to Decision Letter 0]

4 Aug 2020

Response to reviewers

Thank you for giving us the opportunity to submit a revised draft of our study. We sincerely appreciate your effort reviewing our work and have incorporated your valuable comments and suggestions to the revised manuscript. All reference numbers refer to the revised manuscript. 

Response to reviewer #1

Reviewer #1: GENERAL:

This manuscript describes a study in which the repeatability, reproducibility, and agreement of intraocular pressure (IOP) measurement with the Tono-Pen AVIA, a portable McKay-Marg tonometer, using Ocufilm and polyethylene wrap tip covers in human eyes was evaluated.

A gas-sterilized, polyethylene wrap was used as an alternative to the Tono-Pen tip cover as supplied by the manufacturer.

To assess intra-observer repeatability and inter-observer reproducibility, in the right eyes of human subjects, 4 IOP measurements using a polyethylene wrap tip cover were performed by two examiners, A and B, in a randomized manner.

In order to assess intra-observer repeatability and agreement, in the left eyes of human subjects, 4 IOP measurements were performed by two examiners, A and B, using both a polyethylene wrap tip cover and the Ocufilm tip cover in a randomized manner.

Bland-Altman plot and intra-class correlation coefficients (ICC) were used in all analyses. Cost minimization analysis was also performed.

For examiner A, the repeatability of polyethylene wrap tip cover = -0.34, 95% limits of agreement (LOA) = -3.04 to 2.36, and ICC = 0.93 in the right eyes, and -0.33, 95% LOA = -3.01 to 2.36, and ICC = 0.93 in the left eyes.

For examiner B, the repeatability of polyethylene wrap tip cover = - 0.02, 95% LOA = -2.88 to 2.83, and ICC = 0.92. The inter-observer reproducibility of polyethylene wrap tip cover = 0.36, 95% LOA = -3.34 to 4.07, and ICC = 0.95. The repeatability of Ocufilm = - 0.42, 95% LOA =-2.75 to 1.91, and ICC = 0.95.

The authors conclude that for intraocular pressure measurement with the Tono-Pen AVIA, the polyethylene wrap tip cover demonstrates acceptable repeatability, reproducibility, and agreement with the Ocufilm tip cover, and exhibits a good safety profile, at about 1/8 × the cost.

SPECIFIC COMMENTS:

Study Design seems reasonably sound. However a major weakness was, as the authors acknowledge, only ocular normotensive eyes were studied. The essential conclusion of the study, that the Ocufilm versus the polyethylene cover make no significant difference to IOP measurements, may be altered if ocular hypertensive eyes were to be studied. This needs to be addressed, as expounded below.

Answer: We are concerned about the range of IOP that could affect its accuracy because Tono-Pen is known to have less accuracy at higher or lower IOP levels. [28] Thus, our rationale for this study was to prove the use in normal IOP range first before we extend its use to other IOP ranges. 

We have addressed our limitation and edited our conclusion to be more specific and represent the study population.

Limitation: “In addition, our previous study found greater MD and wider LOA in the eye model with higher IOP range compared to the lower ones.[13] Future studies are required to validate the agreement between PF and OF tip cover in eyes with higher IOP range.”

Conclusion: “PW tip cover was safe and demonstrated acceptable intra-observer repeatability, inter-observer reproducibility, with good agreement compared to OF tip cover for IOP measurement with Tono-Pen in normotensive eyes. PW could be used as an alternative tip cover for Tono-Pen.”

Reference

28. Minckler DS, Baerveldt G, Heuer DK, Quillen-Thomas B, Walonker AF, Weiner J. Clinical evaluation of the Oculab Tono-Pen. Am J Ophthalmol. 1987;104(2):168-73.

Reviewer #1: English is quite good but still needs to be redone by a native speaker or professional English writing service. There are too many grammatical errors.

Answer: Thank you for your recommendation. We have our manuscript edited by the English editing service of the Faculty of Medicine, Chulalongkorn University, Thailand. The Certificate of service has been uploaded as a separate file.

Reviewer #1: How was the Tono-Pen AVIA calibrated? Please explain in detail.

Answer: The Tono-Pen AVIA does not require a calibration unless suspect readings are observed, or to ensure that the sensor and electronics are performing as expected. In this study, we calibrated the device every day before use for accurate results. The calibration steps described in the manual are as follows:

1. Hold the Tono-Pen AVIA tonometer with the Transducer Assembly end pointing down towards the floor

2. Press and hold the activation button for 5 seconds – a beep will sound at one second intervals

3. At the end of the 5 second button hold, the display will show [dn]

4. Keep the pen vertical, with the Transducer Assembly pointing down towards the floor, for a total of 15 seconds

5. At the end of this period, a beep will sound and the display will show [up]

6. Immediately point the Transducer Assembly straight up and wait for the next beep (within 3 seconds)

7. A properly functioning Tono-Pen AVIA tonometer will display “PASS”. Pressing the activation button will now put the device into Applanation mode.

The sentence about the Topo-Pen Avia calibration has been modified as follows. “Although regular calibration is not necessary, it was performed per standard protocol recommended by the manufacturer once at the beginning of the day.”

Reviewer #1: In the IOP measurement section, it is stated that ten IOP measurements were taken, and then a mean value was calculated. Was this mean value considered to be a single IOP reading, such that, if 4 IOP values are stated as having been made, does this then imply 4 x 10 = 40 readings, i.e. 40 individual tonometer-corneal contacts?

Answer: The Tono-Pen AVIA requires 10 applanations to obtain a single mean IOP reading (without showing values for each applanation). In clinical practice, we repeat measurements when the IOP reading is questionable. In this study, we performed 40 corneal contacts to obtain 4 IOP readings for each eye.

Reviewer #1: The authors make mention of pseudofacility (i.e. reduction in IOP with repeated tonometric measurement, likely if a large number of individual readings were obtained using this type of device), but make no attempt to indicate by how much IOP may have diminished as a consequence of this. Can they give some indication of this?

Answer: Repeated measurements have shown to lower subsequent IOP readings of Goldmann applanation tonometry. This effect is possible for Tono-Pen as well, but it has not been consistently found in the literature. 

Reference

- AlMubrad TM, Ogbuehi KC. The effect of repeated applanation on subsequent IOP measurements. Clin Exp Optom. 2008 Nov;91(6):524–9.

Reviewer #1: Units (mmHg) should be included on axes of graphs.

Answer: Thank you for your suggestion. We have added units on axes of all graphs.

Reviewer #1: Where is the Legend for Figure 6?

Answer: We are very sorry for the mistake that there were twice of figure 3 in the legends. We have corrected the sequence of figures. Furthermore, we have corrected the ICC value for inter-observer reproducibility and agreement in the revised manuscript.

Reviewer #1: If I am reading it correctly, Figure 6 implies that there was a difference of as much as 4 mmHg between measurements made by Examiner A versus Examiner B. Isn’t this a rather high degree of variation? Or is this simply variation due to pulse amplitude?

Answer: As we cannot directly measure IOP in clinical practice due to the invasiveness, there are several factors affecting IOP measured by any tonometer that cause different variations for each instrument. The variation for Tono-Pen is generally high. Test-retest variability of Tono-Pen was reported up to 3.5 mmHg. [2] Inter-observer reproducibility naturally has higher variation. The difference of as much as 4 mmHg using polyethylene wrap tip cover in this study is acceptable and comparable to our previous studies. [13,14] The repeatability for each examiner is also similar. This variation could not be totally explained by the pulse amplitude because it was also found in our eye model study, in which the pressure was constant. Then, other factors discussed in the manuscript are also significant. 

References

2. Schweier C, Hanson JV, Funk J, Toteberg-Harms M. Repeatability of intraocular pressure measurements with Icare PRO rebound, Tono-Pen AVIA, and Goldmann tonometers in sitting and reclining positions. BMC Ophthalmol. 2013;13:44.

13. Suvannachart P, Asawaphureekorn S. Agreement and repeatability of intraocular pressure measurement between Ocufilm® and plastic wrap TonoPen® tip cover. Paper presented at: The 2nd ASEAN Ophthalmology Society Congress; 2015 Oct 29-31; Hanoi, Vietnam.

14. Suvannachart P, Asawaphureekorn S, Sriphon P, Lertitthikul N, Jitasombuti P, Seesupa S. Safety, repeatability and agreement of a custom-made plastic wrap Tono-Pen® tip cover for intraocular pressure measurement. Paper presented at: The 7th World Glaucoma Congress; 2017 Jun 28-Jul 1; Helsinki, Finland.

Reviewer #1: Again, if I am reading it correctly, Figure 9 implies that there was a difference of as much as 6 mmHg between measurements made with the OF versus the PW tip covers. Isn’t this also a rather a high degree of variation? Or again, is this simply variation due to pulse amplitude?

Answer: Compared to intra-observer repeatability, the variation is generally higher for inter-observer repeatability, and highest for method. For example, agreement between Tono-Pen and Goldmann applanation tonometry showed the mean difference varying from -0.27 to 1.63 mmHg and the range of 95% limits of agreement from 4.88 to 16.34 mmHg in normal subjects [1, 18, 20-23], and the variation was even higher in glaucoma subjects.[24] In our study, both tip covers showed similar repeatability, and the difference of as much as 6 for agreement study is within the range mentioned above and comparable to our previous studies. [13,14] Consequently, we concluded that the difference found in our study was acceptable. This variation could not be completely explained by the pulse amplitude because it was also found in our eye model study, in which the pressure was constant. Then, other factors discussed in the manuscript are also significant. 

References

1. Berk TA, Yang PT, Chan CC. Prospective Comparative Analysis of 4 Different Intraocular Pressure Measurement Techniques and Their Effects on Pressure Readings. J Glaucoma. 2016;25(10):e897-e904.

13. Suvannachart P, Asawaphureekorn S. Agreement and repeatability of intraocular pressure measurement between Ocufilm® and plastic wrap TonoPen® tip cover. Paper presented at: The 2nd ASEAN Ophthalmology Society Congress; 2015 Oct 29-31; Hanoi, Vietnam.

14. Suvannachart P, Asawaphureekorn S, Sriphon P, Lertitthikul N, Jitasombuti P, Seesupa S. Safety, repeatability and agreement of a custom-made plastic wrap Tono-Pen® tip cover for intraocular pressure measurement. Paper presented at: The 7th World Glaucoma Congress; 2017 Jun 28-Jul 1; Helsinki, Finland

18. Nakakura S, Mori E, Yamamoto M, Tsushima Y, Tabuchi H, Kiuchi Y. Intradevice and Interdevice Agreement Between a Rebound Tonometer, Icare PRO, and the Tonopen XL and Kowa Hand-held Applanation Tonometer When Used in the Sitting and Supine Position. J Glaucoma. 2015;24(7):515-21.

20. Schweier C, Hanson JVM, Funk J, Töteberg-Harms M. Repeatability of intraocular pressure measurements with Icare PRO rebound, Tono-Pen AVIA, and Goldmann tonometers in sitting and reclining positions. BMC Ophthalmol. 2013;13:44.

21. Yilmaz I, Altan C, Aygit ED, Alagoz C, Baz O, Ahmet S, et al. Comparison of three methods of tonometry in normal subjects: Goldmann applanation tonometer, non-contact airpuff tonometer, and Tono-Pen XL. Clin Ophthalmol. 2014;8:1069-74.

22. Dey A, David RL, Asokan R, George R. Can Corneal Biomechanical Properties Explain Difference in Tonometric Measurement in Normal Eyes? Optometry and vision science : official publication of the American Academy of Optometry. 2018;95(2):120-8.

23. Kato Y, Nakakura S, Matsuo N, Yoshitomi K, Handa M, Tabuchi H, et al. Agreement among Goldmann applanation tonometer, iCare, and Icare PRO rebound tonometers; non-contact tonometer; and Tonopen XL in healthy elderly subjects. International ophthalmology. 2018;38(2):687-96.

Reviewer #1: As mentioned above, one weakness of the study, as the authors state, is that only ocular normotensive eyes were studied. The Tono-Pen tends to underestimate IOP in ocular hypertensive eyes. It very well might be the case that the different covers (Ocufilm versus polyethylene) could lead to a bigger difference in IOP measurements (depending on how high the true IOP actually is), perhaps too big to be clinically useful. The authors do state this but this needs to be made very clear. Is there any animal data, for example, in their dog study, that might show that the two different covers do not make a significant difference in the tonometric readings at high IOPs? Or any other published data. It is really necessary to show this, prior to using the tonometer with a polyethylene cover in a clinical setting. Perhaps a similar study could be performed in ocular hypertensive patients, after comparing Tono-Pen data with corresponding GAT data.

Answer: As our study introduced the use of new material as a tip cover for Tono-Pen, we decided to first prove our idea with normotensive eyes, which are the majority population. However, in the eye model study, we found that the agreement tended to be lower in the pressure more than 30 mmHg. [13] For the pressure range 10 to 75 mmHg, MD was 1.03 mmHg and 95% LOA were from -6.58 to 8.64 mmHg. In the pressure range 10 to 30 mmHg, MD was 0.29 mmHg and 95% LOA were from -5.68 to 6.26 mmHg. In the future, we plan to perform a similar study in the ocular hypertensive eyes, and possibly compared to GAT value. Thank you very much for your recommendation.

The following sentence has been added to the limitation paragraph. “In addition, our previous study found greater MD and wider LOA in the eye model with higher IOP range compared to the lower ones.[13] Future studies are required to validate the agreement between PF and OF tip cover in eyes with higher IOP range.”

Reference

13. Suvannachart P, Asawaphureekorn S. Agreement and repeatability of intraocular pressure measurement between Ocufilm® and plastic wrap TonoPen® tip cover. Paper presented at: The 2nd ASEAN Ophthalmology Society Congress; 2015 Oct 29-31; Hanoi, Vietnam.

Reviewer #1: When the authors mention their measurements of IOP in dogs in their previous study, did they use the Tono-Pen AVIA? Or the Tono-Pen AVIA Vet (a similar but differently calibrated device, intended for use not with humans but with dogs and cats). Or a different version of the Tono-Pen?

Answer: We also used the Tono-Pen AVIA in the canine study.

Response to reviewer #2

Reviewer #2: Will there be any consideration to adjust stay design to mitigate the risk of Punctate Epithelial Erosions?

Answer: In clinical practice, punctate epithelial erosions commonly occurs after any procedures that require contact on corneal surface, including intraocular pressure measurement with any methods. It is generally asymptomatic and can be resolved spontaneously. However, we recommend users to ensure the smoothness of the film at the instrument tip after insertion for a better precision and less chance of corneal damage. 

Reviewer #2: Line 4: "It demonstrated" changed to "It Demonstrates"

Answer: Thank you for your suggestion. The correction has been made as suggested:

“It demonstrates comparable measurement of IOP compared to the Goldmann applanation tonometry (GAT) which is the gold standard method.”

Reviewer #2: Line 6: Change to "Can be used on patient in any position"

Answer: Thank you for your suggestion. The correction has been made as suggested:

“It can measure the IOP in a small area on the cornea and can be used on patient in any position.”

Reviewer #2: Remove "sometimes happens" from the end of "In addition to shortage at supply"

Answer: Thank you for your suggestion. The sentence was rewritten by the English editing service:

“In addition, sometimes there are shortages of OF tip cover.”

---

## [Decision Letter · Decision Letter 1]

15 Sep 2020

Repeatability, reproducibility, agreement, and safety of Tono-Pen tip cover for intraocular measurement using latex and polyethylene wrap.

PONE-D-20-15592R1

Dear Dr. Pongpirul,

We’re pleased to inform you that your manuscript has been judged scientifically suitable for publication and will be formally accepted for publication once it meets all outstanding technical requirements.

Kind regards,

Sanjoy Bhattacharya

Academic Editor

PLOS ONE

Additional Editor Comments (optional):

Reviewers' comments:

Reviewer's Responses to Questions

**Comments to the Author**

1. If the authors have adequately addressed your comments raised in a previous round of review and you feel that this manuscript is now acceptable for publication, you may indicate that here to bypass the “Comments to the Author” section, enter your conflict of interest statement in the “Confidential to Editor” section, and submit your "Accept" recommendation.

Reviewer #1: All comments have been addressed

Reviewer #2: All comments have been addressed

2. Is the manuscript technically sound, and do the data support the conclusions?

Reviewer #1: Yes

Reviewer #2: Yes

3. Has the statistical analysis been performed appropriately and rigorously? 

Reviewer #1: Yes

Reviewer #2: Yes

4. Have the authors made all data underlying the findings in their manuscript fully available?

Reviewer #1: Yes

Reviewer #2: Yes

5. Is the manuscript presented in an intelligible fashion and written in standard English?

Reviewer #1: Yes

Reviewer #2: Yes

6. Review Comments to the Author

Reviewer #1: (No Response)

Reviewer #2: (No Response)

7. PLOS authors have the option to publish the peer review history of their article (what does this mean?). If published, this will include your full peer review and any attached files.

Reviewer #1: No

Reviewer #2: No

---

## [Editor Report · Acceptance letter]

18 Sep 2020

PONE-D-20-15592R1 

Repeatability, reproducibility, agreement, and safety of Tono-Pen tip cover for intraocular measurement using latex and polyethylene wrap. 

Dear Dr. Pongpirul:

I'm pleased to inform you that your manuscript has been deemed suitable for publication in PLOS ONE. Congratulations! Your manuscript is now with our production department. 

Kind regards, 

on behalf of

Dr. Sanjoy Bhattacharya 

Academic Editor

PLOS ONE